# TURNIP: A "NONDETERMINISTIC" GPU RUNTIME WITH CPU RAM OFFLOAD

## ABSTRACT

An obvious way to alleviate memory difficulties in GPU-based AI computing is via *CPU offload*, where data are moved between GPU and CPU RAM. While CPU offload is useful, it can greatly slow down a computation due to the relatively slow transfer rate between CPU RAM and GPU RAM. To address this, overlapping memory transfer and compute is a necessity, but this asynchronicity introduces non-determinacy, and hence it is impossible to know beforehand what is the best order of operations. We describe TURNIP, which is a system for running AI computations using CPU offload, designed to handle this nondeterminacy. The key innovation in TURNIP is the compilation of the AI computation into a dependency graph that gives the TURNIP runtime freedom to run operations such as GPU kernel calls in many different orders; at runtime, TURNIP chooses the best order on-the-fly, in response to real-time events. We find that TURNIP outperforms standard, PyTorch-based systems supporting constrained GPU RAM by a significant margin, and also avoids out-of-memory errors in a severely memory constrained environment.

## 1 INTRODUCTION

Memory management for modern AI computations is difficult. In the LLaMA large language model (Touvron et al., 2023), for example, the attention computation on a sequence of length $n$ produces an intermediate result with $128 \times n^2$ floating point numbers. Thus, for a long input sequence of 100,000 tokens, the attention computation will result in 1.2 trillion numbers, which would require 2.4 terabytes to store at half-precision. It is for these reasons that out-of-memory errors plague AI programmers (FAQ).

CPU offload—where data are moved to CPU RAM for storage—can help. As CPU RAM is much more inexpensive than GPU RAM and it is possible to install many terabytes of CPU RAM in a GPU server essentially "for free",[1] it makes sense to leverage CPU RAM to temporarily store data. This idea has been explored in several systems such as `pofo` (Beaumont et al., 2021), AutoTM (Hildebrand et al., 2020), SwapAdvisor (Huang et al., 2020), Capuchin (Peng et al., 2020), and POET (Patil et al., 2022). These systems view a GPU computation as a dataflow graph, and plan how to fit the computation into GPU RAM by making use of CPU RAM offload.

While CPU offload is an obvious idea, it can greatly slow down a computation, due to the relatively slow transfer rate between CPU RAM and GPU RAM. Thus, any system for CPU offload must ensure that when such a transfer happens, no computation is blocked waiting for the transfer to finish.

In this paper, we propose TURNIP (short for "nonde**T**erministic gp**U** **R**u**N**time w**I**th c**P**u offload") which is a runtime for multi-GPU servers, designed to systematically support CPU RAM offload. The key innovation of TURNIP is its combination of a pre-computed memory access plan called a MEMGRAPH with a "nondeterministic," event-driven system runtime. A MEMGRAPH is a dependency graph where vertices represent tasks (such as the execution of a GPU kernel to perform a small part of attention computation in a layer of a large language model) and edges represent data or memory dependencies. Any execution order that respects the dependencies in the MEMGRAPH is valid, and tasks are dispatched at any time that their dependencies have been met and the appropriate resources are free. Thus, two executions of the same MEMGRAPH may lead to different sequences of operations

---

[1]At current prices, a single state-of-the-art H100 GPU costs the same as approximately 10TB of CPU RAM—more than $100\times$ the RAM available on a H100 GPU

being executed on a GPU, or different sequences of tensors being paged to CPU RAM—hence the non-determinacy. However, the dependencies in the MEMGRAPH are such that the final output is always *correct*, no matter the execution order. TURNIP's event-driven, fully asynchronous runtime is unique. Because operations can be dispatched whenever the dependencies are fulfilled and are not constrained to any specific ordering, it lowers the chance that any GPU will be stalled waiting for a memory transfer to complete. If one task cannot run due to an un-met dependency in the MEMGRAPH, it is possible that there is another task that *can* run.

The key technical challenge is how to effectively build a MEMGRAPH with as few dependencies as possible, to allow the runtime as much freedom as possible to dispatch operations so that it is never blocked, waiting for a memory transfer to complete. TURNIP builds a MEMGRAPH by simulating an execution of the computation, mapping tensors to GPU memory locations and, when necessary adding edges that represent memory dependencies, as well as `offload` and `reload` operations.

## 2  WHY IS NON-DETERMINACY OF EXECUTION ORDER CRUCIAL?

The design of TURNIP is based on a simple hypothesis: When running a GPU-based computation that utilizes CPU RAM, asynchronous operations such as `offload` and `reload` will have a seemingly nondeterministic running time that is difficult to pre-plan for. The system runtime must accommodate the resulting non-determinacy, or else performance can suffer.

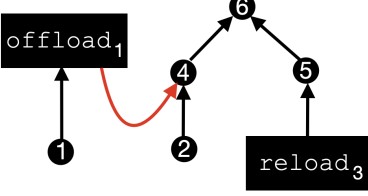

Consider Figure 1, which depicts a MEMGRAPH for a single GPU system with an `offload` (data movement from GPU RAM to CPU RAM) and a `reload` (from CPU RAM to GPU RAM). Vertices are operations (for example, GPU kernel calls) that produce data. Black edges indicate data or consumption dependencies; red edges indicate memory dependencies (MEMGRAPHs will be described in detail in Section 4). Note the memory dependency from `offload₁` to **4**. This exists because the output of **4** will be written to the location of the output of **1**, and so **4** cannot execute until the `offload₁` completes. The data dependency from `reload₃` to **5** exists because the kernel associated with vertex **5** will consume the `reload`ed tensor.

Figure 1: A simple MEMGRAPH.

Imagine that a system has executed GPU kernels associated with vertices **1** and **2**. It is currently executing the `offload₁` and the `reload₃`. At this point, *it is impossible to know which kernel should run next* (**4** or **5**) as this depends on which memory transfer finishes first. Ideally, *this decision will be made at runtime*. If the system deterministically decides to run **4** before **5** at compile time, and the `reload₃` finishes first, the GPU will sit, idle, waiting for the `offload₁` to complete. This is why a special-purpose, "nondeterministic" runtime is needed: to properly handle the non-determinacy induced via the addition of memory operations.

## 3  RELATED WORK

There are two approaches taken by systems dealing with limited GPU memory. Some, like TURNIP, accept an abstracted version of a generic GPU computation. Other systems are more specifically targeted to certain categories of models, optimization algorithms, or to specific tasks such as training or inference. Unlike TURNIP, none of these existing systems consider the effect of the non-determinism of `offload` and `reload` operations on system performance, nor do any focus on the system runtime.

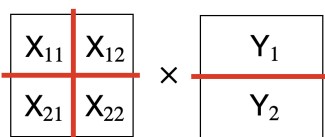

Using the first, more general approach, are systems that accept a generic dataflow graph and, like TURNIP, plan for execution in limited memory: `pofo` (Beaumont et al., 2021), AutoTM (Hildebrand et al., 2020), SwapAdvisor (Huang et al., 2020), Checkmate (Jain et al., 2020), Capuchin (Peng et al., 2020), and POET (Patil et al., 2022) all assume an input dataflow graph for a machine learning computation, and then plan for execution in limited memory. Checkmate considers only tensor re-materialization, whereas

Figure 2: A decomposition of matrix multiplication.

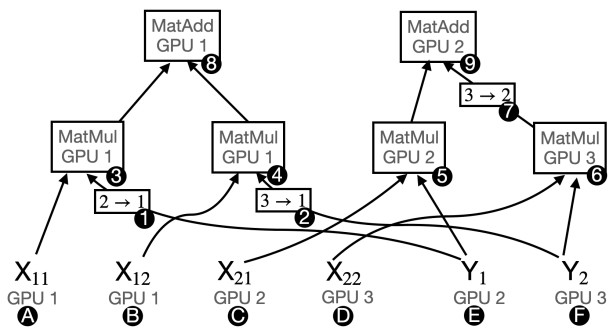

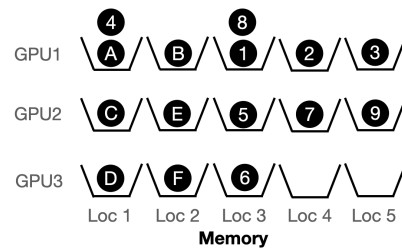

Figure 3: Example TASKGRAPH consisting of six GPU kernel calls and three GPU-to-GPU transfers.

Figure 4: Possible mapping of the output of all of the operations in Figure 3 to memory locations.

POET, pofo, and Capuchin consider re-materialization and offload; AutoTM and SwapAdvisor consider only offload.

The more targeted approach is taken by the DeepSpeed project (Deepspeed) and the various ZeRO optimizations. For transformers and other, similar models, DeepSpeed inference (which includes ZeRO-Inference) (Aminabadi et al., 2022) has two key ideas. First, DeepSpeed inference "offload[s] some activation from GPU to CPU memory while not in use." Second, DeepSpeed inference "pins the model weights either in DRAM (if large enough) or NVMe, and streams each layer into GPU memory for computation when needed." FlexGen (Sheng et al., 2023) seeks to use a variety of methods to speed transformer inference given limited hardware, including model weight offload to CPU, quantization (Yao et al., 2022; Frantar et al., 2022), and sparse attention (Child et al., 2019). The latter two ideas are orthogonal to the ideas in this paper. For CPU offload, FlexGen optimizes a "zig-zag" block scheduling that works through transformer layers and sequences in the batch, offloading and reloading the KV-cache (Pope et al., 2023) and model weights. PagedAttention (Kwon et al., 2023) deals with low memory utilization in transformers, developing a paging system for the KV-cache.

ZeRO-Offload (Ren et al., 2021) is a comprehensive solution for limited-memory training that can be seen as primarily using CPU RAM for running the ADAM optimizer, moving weights to GPU RAM on a carefully-controlled schedule. ZeRO-Offload is an enhancement on ZeRO (Rajbhandari et al., 2020), which is designed to be memory-efficient, partitioning both the optimizer and the data across multiple GPUs. ZeRO-Infinity (Rajbhandari et al., 2021) is similar, and includes a CPU offload engine, as well as tiling of operators to utilize the RAM of multiple GPUs.

## 4    TASKGRAPHS AND MEMGRAPHS IN TURNIP

TURNIP takes is input a TASKGRAPH. A TASKGRAPH is a dataflow graph (a directed, acyclic graph) that describes how to perform multi-GPU computations. In a TASKGRAPH, edges represent data flow, and vertices represent operations over tensors. A vertex without any inputs (called an *input vertex*) is associated with an input tensor. An operation associated with a non-input vertex may be either a kernel call that is to be executed on a specific GPU, or a GPU-to-GPU data transfer.

TURNIP is agnostic as to how the TASKGRAPH is created; it could, for example be created using a framework such as FlexFlow (Jia et al., 2019) or Alpa (Zheng et al., 2022). Consider a matrix multiplication $\mathbf{X} \times \mathbf{Y}$, and assume we wish to execute this matrix multiplication on three GPUs. To produce a TASKGRAPH, a framework such as FlexFlow may choose to decompose this matrix multiplication as depicted in Figure 2, perhaps corresponding to the TASKGRAPH of Figure 3.

Given such a TASKGRAPH, TURNIP first compiles the TASKGRAPH into a MEMGRAPH, which it will eventually execute. Like a TASKGRAPH, a MEMGRAPH is also a directed acyclic graph. Every vertex in the original TASKGRAPH will be present in a corresponding MEMGRAPH. Further, the compilation process may add additional offload and reload operations that move memory from GPU RAM to CPU RAM, and vice versa. During the compilation process, the output associated with every vertex in the MEMGRAPH is mapped to a memory location. Unlike the input TASKGRAPH, the MEMGRAPH

is not a dataflow graph; it is a dependency graph. If there is an edge from $v_1$ to $v_2$, it means that $v_2$ depends on $v_1$ and $v_2$ may not execute until after $v_1$ has been executed. In a MEMGRAPH, there are two types of dependencies. One is a data dependency, which is inherited from the TASKGRAPH (or is created via the addition of an `offload` or `reload`; see below). The second is a memory dependency, which is added to ensure that there are no race conditions in the graph. A *race condition* occurs when there is some vertex for which two valid executions of the graph may produce a different output. This can happen when two vertices write to the same memory location, and it is possible for a third vertex to read either output, depending upon the execution order.

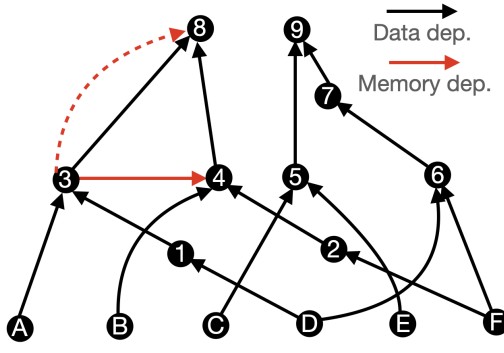

Figure 5: MEMGRAPH corresponding to Figure 3.

Let us illustrate a possible compilation of the TASKGRAPH of Figure 3 to a MEMGRAPH. Imagine that our three GPUs each have five memory locations, and for simplicity, each tensor is the same size and occupies exactly one memory location. During compilation, the tensor associated with the output of each operation in the TASKGRAPH is assigned to a memory location, as depicted in Figure 4. GPU 1 must deal with seven tensors total (two input tensors and five additional tensors that are created via the execution of some operation), and we cannot fit all seven of those tensors in memory, given our five locations. Thus, the tensors output by operations A and 4 are both mapped to GPU1-Loc1, and the tensors output by operations 1 and 8 are both mapped to GPU1-Loc3.

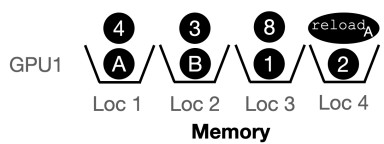

Figure 6: Possible mapping of tensors to GPU RAM.

A corresponding MEMGRAPH is shown in Figure 5. Note that two new edges representing memory dependencies have been added. These edges guarantee that the graph is free of race conditions. Specifically, a graph will be free of race conditions, if, whenever the outputs of vertices $v_1$ and $v_2$ have both been mapped to the same memory location, either $v_1$ *safely overwrites the result of* $v_2$, or $v_2$ *safely overwrites the result of* $v_1$. We say that "$v_1$ safely over-writes the result of $v_2$" if and only if, for every $v_3$ that consumes the output of $v_2$, there is a memory dependency from $v_3$ (or some descendent of $v_3$) to $v_1$ (or to some ancestor of $v_1$). Why? If $v_1$ is to safely over-write the result of $v_2$, we need to ensure that $v_1$ cannot execute until all of the consumers of $v_2$ have finished execution—such memory dependencies ensure this.

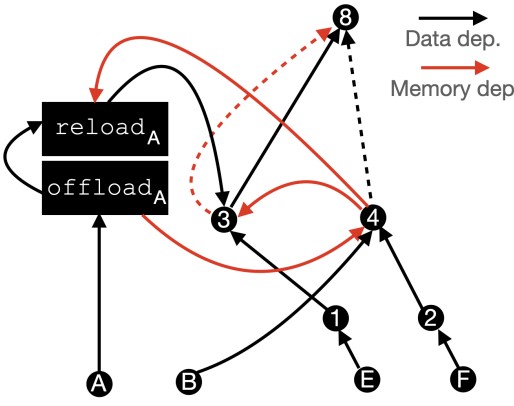

Figure 7: MEMGRAPH with less GPU RAM.

For example, from Figure 4 we see that the output of vertex 4 is mapped to the same location as the output of vertex A. In the associated MEMGRAPH of Figure 3, to ensure that 4 safely over-writes the result of A, we add a memory dependency from 3 (the only consumer of A) to 4. From Figure 4 we also see that the results of 1 and 8 are mapped to the same location. To ensure that 8 safely over-writes the result of 1, we add a memory dependency from 3 (the only consumer of 1) to 8. Note that this memory dependency is shown as a dashed line; this indicates that it is *superfluous*, as there is already a data dependency from 3 to 8, so this memory dependency is not needed for correctness.

Things can become more intricate if the memory is more constrained. Consider the case where we have only four memory locations on each GPU, and we wish to compile the same TASKGRAPH. One possible mapping of the vertices TASKGRAPH of Figure 3 to memory locations for GPU 1 is shown in Figure 6; the associated TASKGRAPH is shown in Figure 7. Note in particular the addition of an `offload`-`reload` pair. Both the `offload` and the `reload` are new operations that are

added to the MEMGRAPH during compilation, to facilitate execution in memory-constrained scenarios. We can always compile $v_1 \to v_2$ in a TASKGRAPH to $v_1 \to \text{offload}_{v_1} \to \text{reload}_{v_1} \to v_2$ in a MEMGRAPH. After the $\text{offload}_{v_1}$, the result of $v_1$ takes up no GPU memory, but it cannot be used until the $\text{reload}_{v_1}$, where it is again mapped to a GPU memory location. The reason for the inclusion of the offload-reload pair in this case is that it allows the result of A to be removed from GPU RAM for a time. Thus, vertex 4 can execute and write its result on top of the result of A, which is subsequently reloaded so that vertex 3 can be executed.

In Figure 6 we see that there are four pairs of vertices whose results are mapped to the same GPU memory locations, and so memory dependencies must been added to the MEMGRAPH to ensure that there are no race conditions. Consider A and 4, which are both mapped to GPU1-Loc1. To ensure that 4 safely over-writes the result of A, we have a memory dependency from the $\text{offload}_{v_1}$ (the only consumer of A) to 4. Or, consider $\text{reload}_{v_1}$ and 2, which are both mapped to GPU1-Loc4. To ensure that the $\text{reload}_{v_1}$ safely over-writes the result of 2, there is a memory dependency from the only consumer of 2 (vertex 4) to the $\text{reload}_{v_1}$.

## 5 THE TURNIP EXECUTION ENGINE

Once a MEMGRAPH has been produced, it is executed by the TURNIP engine using a nondeterministic, event-based framework. As soon as a GPU is unused or a tensor is ready to be offloaded to RAM, the TURNIP runtime can immediately assign any available work to the GPU or begin the transfer, without regard to the overall state of the computation. Also note that there are no calls to memory-management routines such as cudaMalloc or cudaFree during MEMGRAPH execution, as memory management is no longer dynamic. Tensor placement is pre-determined before execution, and if dependencies are respected, there can be no memory corruption due to race conditions.

To execute the MEMGRAPH, TURNIP runs a central event processing loop, that repeatedly processes callback functions that are called response to completion of the work associated with a MEMGRAPH vertex (completion of a GPU-to-GPU transfer, completion of the GPU kernel, or completion of an offload or reload). When a vertex completes and a callback is invoked, the event loop checks to see if any other vertex can be executed. That is, it searches for a vertex $v_1$ where (a) all vertices $v_2$ with an edge $v_2 \to v_1$ in the MEMGRAPH have also completed; (b) if $v_1$ is a kernel call, then the GPU $v_1$ is assigned is currently free. When the event loop finds such a vertex, it launches it, and searches for another such vertex. When it can find no such executable vertex, it goes to sleep until woken by another callback.

## 6 BUILDING A MEMGRAPH

The key technical question we address in this paper is: How to construct a MEMGRAPH from a TASKGRAPH? The primary requirement for the compilation process is *correctness*. Correctness requires that (a) every data dependency present in the TASKGRAPH is also present in the MEMGRAPH, *or* is replaced with a sequence of offload-reload operations;[2] (b) there are no race conditions in the MEMGRAPH; (c) the MEMGRAPH has no cycles. In addition, it is desirable for the MEMGRAPH to be performant. A MEMGRAPH will not be performant if memory dependencies severely constrain the execution order of vertices. Such constraints may reduce parallelism and GPU utilization.

Our basic tactic during compilation is to rely on a simulated execution of the TASKGRAPH to generate the MEMGRAPH. Given a serial ordering of the vertices in the TASKGRAPH that respects all dependencies (so that if $v_1 \to v_2$ is in the TASKGRAPH, $v_1$ is before $v_2$ in the ordering) we simulate its execution, making calls to special variants of malloc and free that do not actually allocate GPU RAM, but instead maintain a map of used and free RAM slots on the GPU that is the target of the compilation. These implementations also maintain a history of which tensors occupied which positions in simulated GPU RAM, to correctly generate memory dependencies. As the simulation runs, the MEMGRAPH is constructed. Calls to the special malloc implementations associate MEMGRAPH vertex outputs to GPU memory locations (effectively producing the mappings depicted in Figure 4 and Figure 6). Whenever a call to the malloc variant fails because there is

---

[2]So, for example, if $v_1 \to v_2$ is present in the TASKGRAPH, we may have $v_1 \to \text{offload}_{v_1} \to \text{reload}_{v_1} \to v_2$ in the MEMGRAPH

not enough GPU RAM, an `offload` vertex must be added to the MEMGRAPH. Whenever it is time to simulate the execution of a TASKGRAPH vertex but one of the inputs is not in the simulated GPU RAM, then a memory location for the corresponding `reload` vertex is allocated, and a data dependency on that `reload` is added to the MEMGRAPH.

As the simulation runs, there are two *horizons*, or counters that mark progress through the serialized TASKGRAPH. The first is the `allocHzn`. Every vertex in the TASKGRAPH that is older than the `allocHzn` has had a space allocated for it. The second is the `execHzn`. Every vertex in the TASKGRAPH that is older than the `execHzn` has been "run" according to the simulation. To ensure a high-quality MEMGRAPH, our compilation algorithm greedily tries to push the `allocHzn` as far as possible past the `execHzn`. Intuitively, this will produce fewer constraints in the resulting MEMGRAPH. A kernel associated with a vertex cannot run until it has GPU RAM to write its output. If this GPU RAM is available very early in the simulation, then it gives the TURNIP event processing loop more freedom to choose a vertex execution order that does not exactly match the simulated ordering, generating more opportunities to run available kernels while waiting for memory transfers.

The overall algorithm, BUILDMEM-GRAPH, is given above in Figure 8. Note that this variant of the algorithm assumes each tensor takes up exactly one slot in GPU RAM. In the "real life" case where tensors are variably-sized, the algorithm does not change appreciably—specifically, in the variably-sized case, freeing space for a tensor can evict a variable number of tensors to CPU RAM—but assuming uniformly-sized tensors simplifies the presentation.

BUILDMEMGRAPH: **Inputs**: TASKGRAPH, sorted list of TASKGRAPH vertices $V = \langle v_1, v_2, ..., v_n \rangle$; **Outputs**: MEM-GRAPH, GPU memory location $v_i.\texttt{loc}$ for $i \in \{1...n\}$

```
Evicted ← {}; execHzn ← 1; allocHzn ← 1;
while execHzn ≤ n do
  if allocHzn <= n and (v_allocHzn.loc ←
  simMalloc(v_allocHzn)) ≠ −1 then
    /*successfully allocated space for future result*/
    allocHzn += 1
  else if allocHzn = execHzn then
    /* unable to allocate for next execution w/o evict*/
    v_allocHzn.loc ← simMallocOffld(v_allocHzn)
    allocHzn += 1
  else
    /* simulate execution of the next vertex */
    /* first, compute set of vertices exec depends on */
    Deps ← {v s.t. edge v → v_execHzn ∈ TASKGRAPH}

    for v ∈ Deps do
      /* reload dependency if evicted */
      if v ∈ Evicted then
        v.loc ← simMallocForceReld(v)
      end if
      /* if dependency won't be used again, free it */
      if not ∃(fut > execHzn s.t.  edge v →
      v_fut ∈ TASKGRAPH) then
        simFree(v)
      end if
      add edge v → v_execHzn to MEMGRAPH
    end for
    execHzn += 1
  end if
end while
```

Figure 8: Building a MEMGRAPH via execution simulation.

At the highest level, the algorithm operates by first checking to see if it can allocate space for the vertex at the current allocation horizon, $v_{\texttt{allocHzn}}$. If it cannot, the algorithm makes sure there is space available for the output of the next vertex to be executed (the only way there is no space is if `allocHzn = execHzn` and the last allocation failed; this implies it is time to execute $v_{\texttt{execHzn}}$ and we just failed to allocate space for it). If there is space, the simulation "executes" $v_{\texttt{execHzn}}$.

There are four memory management subroutines used by the algorithm: three variants on `malloc` (`simMalloc`, `simMallocForceReld`, and `simMallocOffld`) and one variant on `free` called `simFree`. Like a traditional `malloc`, `simMalloc` finds an open slot for the allocation, but it also adds the memory dependencies to the MEMGRAPH necessary to ensure that the vertex $v$ that will occupy the slot will safely overwrite the previous occupant of the slot. `simMallocForceReld` is like `simMalloc`, but it is used in the case when a vertex must be reloaded because it is going to be used immediately, and hence the allocation cannot fail. `simMallocOffld` is a variant of `simMalloc` that cannot fail, as it finds a victim to offload to ensure the success of the allocation for vertex $v$, adding the `offload-reload` sequence to the MEMGRAPH. Crucially, it renames all

instances of the victim $v'$ in the TASKGRAPH to refer to $\texttt{reload}_{v'}$. In this way, all "future" accesses to $v'$ will refer, in fact, to its reloaded version. The routine also adds a memory dependency from the $\texttt{offload}_{v'}$ to $v$, as we cannot execute $v$ until the $\texttt{offload}_{v'}$ has taken place, and freeing GPU RAM for use.

---

simMalloc: **Input**: vertex $v$; **Output**: GPU memory slot for $v$

---

**find** open $\texttt{slot}$ for $v$; **return** -1 if none
**return** $\texttt{slot}$ if no previous occupant
$v' \leftarrow$ last owner of slot for $v$
Deps $\leftarrow$ $\{v'' \text{ s.t. edge } v' \rightarrow v'' \in$ TASKGRAPH$\}$
**for** $v \in$ Deps **do**
   **add** edge $v'' \rightarrow v$ to MEMGRAPH
**end for**
**return** slot

---

simMallocForceReld: **Input**: vertex $v$; **Output**: GPU memory slot for $v$

---

**remove** $v$ from Evicted
$\texttt{slot} \leftarrow \texttt{simMalloc}(v)$
**if** $\texttt{slot} \neq -1$ **then**
   **return** $\texttt{slot}$
**end if**
**return** $\texttt{simMallocOffld}(v)$

---

simMallocOffld: **Input**: vertex $v$; **Output**: GPU memory slot for $v$

---

**find** GPU RAM $\texttt{slot}$ for $v$ and determine victim (current occupant of $\texttt{slot}$) $v'$
**add** sequence $v' \rightarrow \texttt{offload}_{v'} \rightarrow \texttt{reload}_{v'}$ to MEMGRAPH
**add** edge $\texttt{offload}_{v'} \rightarrow v$ to MEMGRAPH
Deps $\leftarrow$ $\{v'' \text{ s.t. edge } v' \rightarrow v'' \in$ TASKGRAPH and $v''$ comes before $v$ in $V\}$
**for** $v \in$ Deps **do**
   **add** edge $v'' \rightarrow v$ to MEMGRAPH
**end for**
**rename** all instances of $v'$ in TASKGRAPH to $\texttt{reload}_{v'}$
**add** $\texttt{reload}_{v'}$ to Evicted
**return** $\texttt{slot}$

---

Figure 9: simMalloc variants used in MEM-GRAPH construction.

**Ensuring the "Unobtrusiveness" of Dependencies.** Even under nondeterministic execution, memory dependencies can cause periods of time where a GPU is not used, as the GPU is blocked on a memory transfer or on a dependency related to two tensors being mapped to the same location in GPU RAM. The likelihood that dependencies added to the MEMGRAPH will cause such stalls can be reduced by a careful implementation of the various $\texttt{malloc}$ variants. For example, when $\texttt{simMallocOffld}$ searches for a victim, we search for the victim whose next use is furthest in the future—that is, closest to the end of $V$ (in the general case where tensors have different sizes and there may be more than one victim, we seek to maximize the minimum age of any evicted tensor). When $\texttt{simMalloc}$ finds an open slot for a tensor, it should choose the available slot whose last use was furthest in the past (closest to the beginning of $V$).

# 7 CORRECTNESS
## OF MEMGRAPH CONSTRUCTION

By construction, all edges present in the TASK-GRAPH are present in the MEMGRAPH, as when a vertex is "executed" during the simulation, all of its incoming data dependencies are added to the MEMGRAPH.

To ensure correctness with respect to race conditions and the absence of cycles, the algorithm relies on the total ordering of vertices in $V$, the list input into BUILDMEMGRAPH. Further, all new $\texttt{offload}$ and $\texttt{reload}$ vertices produced by the compilation process also have a consistent placement in this ordering. Consider the case when $\texttt{simMallocOffld}$ or $\texttt{simMallocForceReld}$ are called to obtain data necessary to execute vertex $v$, and an $\texttt{offload}$ or a $\texttt{reload}$ is produced. All of those $\texttt{offloads}$ and $\texttt{reloads}$ take place just before $v$, with all of the $\texttt{offloads}$ happening just before the $\texttt{reloads}$.

This ordering ensures that there can be no race conditions in the output MEMGRAPH: if the outputs of $v_1$ and $v_2$ are mapped to the same memory location and $v_1$ comes before $v_2$ in the ordering, then the BUILDMEMGRAPH algorithm ensures that $v_2$ safely overwrites the result of $v_1$. Consider the implementation of $\texttt{simMalloc}$. Whenever a tensor produced by $v$ is mapped to a memory slot previously occupied by the result of $v'$, we add edges to ensure that every consumer of $v'$ executes before $v$. Also, consider $\texttt{simMallocOffld}$, where $v'$ is offloaded to CPU RAM to accommodate $v$. Here, memory dependencies are added from the $\texttt{offload}_{v'}$ to $v$ and from all consumers of $v'$ to $v$ (when those consumers appeared before $v$ in the list $V$). Note that the TASKGRAPH is modified

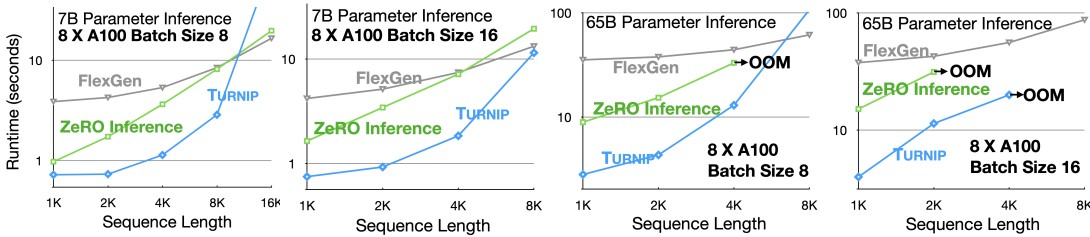

Figure 10: Time for LLaMA first token (prefill) inference, A100 server. "OOM" is out-of-memory.

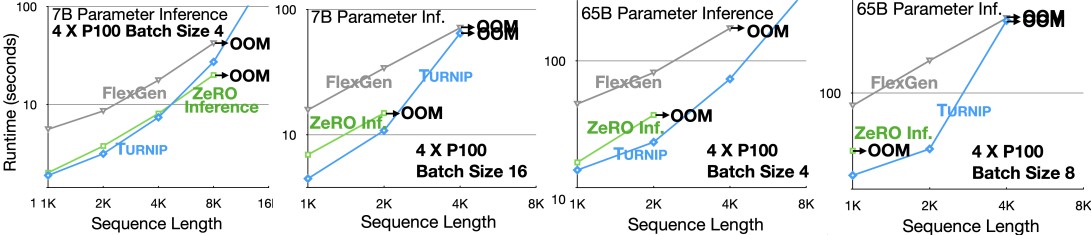

Figure 11: Time for LLaMA first token (prefill) inference, P100 server. "OOM" is out-of-memory.

so that all "future" consumers of $v'$ will consume $\texttt{reload}_{v'}$ rather than $v'$, so they cannot induce a race condition over $v$ and $v'$.

Also, consider why there can be no cycles in the output MEMGRAPH: all edges added to the MEMGRAPH point forward in the total ordering. Consider the edges added by $\texttt{simMalloc}$. $v$ can only be mapped to the location used by $v'$ if $v'$ has been previously $\texttt{free}$'ed. This implies that any vertices using the output of $v'$ have already been "executed", and so come before $v$ in the total ordering. Thus, any edge from a consumer of $v'$ to $v$ must point forward in $V$. Also consider the edges added by $\texttt{simMallocOffld}$. A similar argument holds here, as we explicitly only add edges that point forward in $V$.

## 8 EXPERIMENTS

Our experiments evaluate the ability of TURNIP to deal with Meta AI's LLaMA large language model (LLM) (Touvron et al., 2023), with severely constrained memory. LLM training and inference are chosen as representative, challenging computational workloads encountered by modern ML systems, particularly difficult given the large memory footprint. We assessed TURNIP's performance on 7 billion and 65 billion parameter models.

The system is implemented in C++, with most GPU kernels generated using Nvidia cuTensor. Experiments were conducted on two machines: (i) an older $4 \times$ P100 GPU server (16 GB RAM each) and 22, 64GB DDR4 2666MHz CPU RAM modules, for a total of 1.3TB of RAM, and (ii) an Amazon Web Services $\texttt{p4d.24xlarge}$ instance, equipped with eight A100 GPUs (40 GB RAM each) and 1.15TB of RAM. We were particularly interested in seeing the ability of TURNIP to operate in a difficult environment with extremely limited GPU RAM, hence the P100 GPUs, with only 64 GB of GPU RAM total on the server. Key quesitons are: Can software help bridge the gap—particularly the lack of GPU RAM—between older and newer hardware? Can TURNIP facilitate model training and inference in a situation with limited RAM?

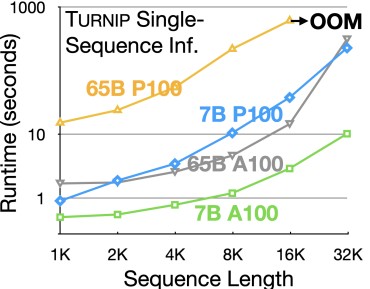

Figure 12: Single-sequence inference times.

**(1) LLaMA first token inference.** Our experiments target "first token" inference (also known as "prefill"): How long does it take to produce the first output token, given an input prompt? We focus on prefill as it is exceedingly expensive in terms of the

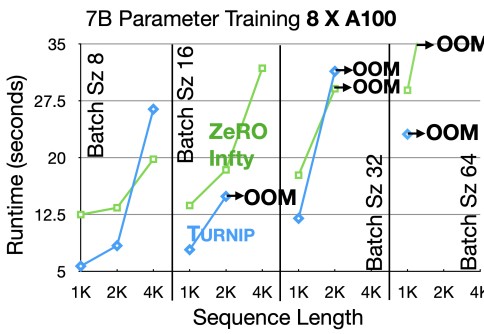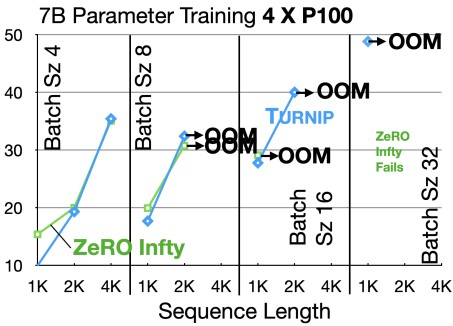

Figure 13: Comparing TURNIP and ZeRO Infinity for LoRA training.

memory required, scaling quadratically with the size of the prompt. On both machines, we run TURNIP, ZeRO Inference (Aminabadi et al., 2022) (using weight partitioning and model weight offload), and FlexGen (Sheng et al., 2023). Note that these are PyTorch-based systems, whereas TURNIP is not. For FlexGen, we use full CPU offload for activations. All testing is done using batched input, as batching is required for FlexGen and ZeRO (as TURNIP simply runs a dataflow graph, it is agnostic to batching). For the smallest batch sizes considered, we test prefill input sequence lengths: 1K, 2K, 4K, 8K, and 16K tokens. For larger batches we use 1K, 2K, 4K and 8K. For TURNIP, all model weights and computations were performed using 16-bit floating points, though FlexGen uses very low precision arithmetic to save RAM and speed compute. Results for the A100 GPU server are given in Figure 10. Results for the P100 GPU server are Given in Figure 11.

One of the advantages of TURNIP is that it executes arbitrary dataflow graphs in limited memory. Thus, as long as a computation is appropriately decomposed to run on multiple GPUs, TURNIP can execute it. This means, for example, that TURNIP does not need to perform inference over batches of input sequences and supports arbitrary combinations of model and data parallelism (unlike FlexGen and in ZeRO Inference). While batching tends to increase computational efficiency, the RAM used by a large batch means it is not possible to run inference over long sequences in limited memory (batching precludes that all 320GB of GPU RAM on be dedicated to prefill for a single long sequence). To investigate the ability of TURNIP to perform inference over a single long sequence, we test sequences sizes of up to 32K tokens, on both GPU servers and on both the 7B and 65B parameter models. Results are shown in Figure 12.

**(2) LoRA training for LLaMA.** We also experiment with LoRA training (Hu et al., 2021). We use a LoRA rank of 16, and train LoRA adaptors for the $K$, $V$, $Q$, and feedforward mapping matrices. Here we run TURNIP and ZeRO Infinity (Rajbhandari et al., 2021); ZeRO is executed using all three "stages" (gradient partitioning, model weight partitioning, optimizer state partitioning) as well as CPU offload. Both TURNIP and ZeRO use checkpointing during the forward pass to reduce the memory footprint. We measure the time it takes to run the forward and backward pass for one batch, with varying batch sizes and sequence lengths. All TURNIP model weights are stored as single precision (32 bits). Results for training using both the P100 and the A100

| Experiment | Avg. Speedup |
|---|---|
| A100 Inference | 4.02% |
| P100 Inference | 6.45% |
| A100 Training | 13.4% |
| P100 Training | 14.5% |

Figure 14: Observed speedup due to nondeterministic ordering, compared to (partially) deterministic ordering.

server at 7B parameters are given in Figure 13. Both systems had a difficult time training the 65B parameter model. TURNIP was faster for the one case it was able to run (1K length sequence, batch size eight took 58.5 seconds using TURNIP and 72.9 using ZeRO Infinity) but Zero Infinity was more robust to larger batch sizes, where TURNIP failed.

**(3) The effect of nondeterminism.** One of the key hypotheses of this paper is that nondeterminism can help performance, by allowing the system to react to the observed state of the computation. To test this, we implement a (semi-)deterministic version of TURNIP and run the same set of experiments using this new version. Specifically, we add dependencies to fix the order of operations at each of the GPUs to be exactly the order that is input into BUILDMEMGRAPH. Unfortunately, this does not

remove all nondeterminism from the system because of the way that reductions (summations) of tensors are handled by the system; fully removing all nondeterminism from TURNIP is not easily possible. Still the experiment may be instructive. Figure 14 shows the speedup obtained by the nondeterministic version compared to the deterministic, over all experiments.

**Discussion.** Throughout the first token inference experiments, TURNIP typically performed the best in terms of latency, with ZeRO Inference generally much slower but outperforming FlexGen, except for larger sequence sizes. This would seem to validate the highly non-deterministic, dataflow-based approach advocated for in the paper, at least if the goal is low latency.

To be fair, we note that FlexGen is designed for high throughput, as opposed to low latency, and FlexGen utilizes multiple GPUs only via pipelined parallelism. Note that FlexGen does not seem to get any slower when moving from batch size of eight to 16 on the A100 server. This suggests that filling the pipeline leads to substantial latency. Further, pipelined parallelism is more effective with more work in each pipeline stage, due to the high synchronization overhead and the need to try to overlap communication with computation, perhaps explaining FlexGen's better performance for larger sequences, which are more dense computationally.

ZeRO Inference takes a much different approach, but it uses a highly synchronized form of model parallelism as it traverses the levels in a transformer, also carefully trying to overlap communication and computation, which may simply be more effective when there is more work to at each level. TURNIP, on the other hand, is radically different. It does not "understand" the levels in a transformer, does not need to synchronize processing of the various levels, and simply tries to asynchronously process kernels as fast as it can. If it is stuck waiting for communication, it simply tries to do something else.

For training, there were clear advantages of TURNIP over ZeRO Infinity, especially for smaller sequence lengths. This was particularly true on the A100 server, where TURNIP was often much faster. For batches of sequences of length 1K, TURNIP often took less than 50% of the time to process each batch, compared to ZeRO (at a batch size of 8, the time to process 1K sequences was 5.6 seconds for TURNIP and 12.5 seconds for ZeRO, for batch size of 32 it was 12.1 seconds for TURNIP and 17.8 seconds for ZeRO). The differences in performance were much less pronounced on the P100 server, though there TURNIP was still faster. Finally, we note that both systems suffered significant out-of-memory errors during training. Interestingly, TURNIP seemed to have more problems with memory on the A100 server, whereas ZeRO Infinity had more problems with memory on the P100 server. We conjecture that some of that could be solved in TURNIP with a better input dataflow graph, which cuts the input problem into smaller pieces.

## 9 CONCLUSIONS

We present TURNIP, a system and runtime for executing memory-intensive computations on GPU servers, supporting CPU offload. The key innovation of TURNIP is its reliance on a "nondeterministic" runtime where a dependency graph is used to guide execution of the computation; any order of operations is acceptable, if the dependencies are respected. We argue that this is necessary when CPU RAM offload is used, or else the system will often be stalled, waiting for CPU-to-GPU transfers.

**Limitations.** As currently implemented, the biggest limitation of TURNIP is that the input computation (in the form of a TASKGRAPH) must be static, and known completely beforehand, so that the MEMGRAPH can be constructed. This is not of consequence during model training, but can be an issue during any recursive, generative AI computation. This includes LLM inference, where the next token must repeatedly be generated and the $KV$-cache increases in size. There are some naive solutions to this (such as pre-compiling a MEMGRAPH for a specific number of token generations in the case of an LLM) but more work will be necessary to create a satisfactory solution to recursive computation.

Another limitation is that while our experiments did show that TURNIP has certain performance advantages, it is not possible to be absolutely sure where those advantages come from. Our ablation shows that a fixed execution order slows down TURNIP, suggesting that nondeterminism is important. But unlike ZeRO and FlexGen, TURNIP was implemented from the ground up and does not rely on Python or PyTorch—that fact alone could account for some of the performance differences.

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

## A    TURNIP ENGINE DETAILS

Our execution engine consists of a central event loop that "launches" each vertex in the MEMGRAPH. A vertex can be launched when (1) all dependencies have been completed and (2) the required resources are obtained. When a vertex is launched, the corresponding operation is executed and then a provided callback is called to notify the event loop that the vertex has completed. In turn, the event loop frees up the obtained resources and keeps track of when vertices complete execution so that subsequent vertices can be launched. In practice, when launched, a vertex will execute one or more asynchronous CUDA operations on CUDA stream and will then call `cudaStreamAddCallback`. As such, every vertex requires as a resource a stream, where a single stream can only be used by a single launched vertex at a time. We use 5 streams per GPU. `offload`, `Reload` and inter-GPU communication vertices will call `cudaMemcpyAsync`. For CPU storage, we allocate a single, large contiguous block of memory with `cudaHostAlloc` with flags `cudaHostAllocPortable` and `cudaHostAllocWriteCombined`. When executing `Offload` vertices, we allocate into the CPU storage memory using our custom allocator; when executing `Reload` vertices, we free from our custom allocator. All compute vertices are executed using either cuTensor functions or hand-written CUDA kernels. An example where a hand-written CUDA kernel is beneficial is for executing portions of softmax so that less workspace memory and fewer vertices would be required. Two additional resources may be required for computing vertices: workspace memory as required for executing multiple cuTensor functions and locks around write-protected memory. As an example, we would execute a summation of $n$ tensors with $n$ calls to tensor increment sum-into kernels. However, the output memory would be protected by a resource so that only one sum-into can happen at a time. This implementation is designed to support non-determinism. We use CUDA version 11.8.0 and cuTensor version 2.0.1. All other code is C++.

