# OpenReview forum: "TURNIP: A “Nondeterministic” GPU Runtime with CPU RAM Offload"
_ICLR.cc/2025/Conference — Submitted to ICLR 2025_

### Official Review · Reviewer_mXjm · 2024-10-29

**Soundness:** 3
**Presentation:** 2
**Contribution:** 2
**Rating:** 5
**Confidence:** 3

**Summary:**

The paper presents TURNIP, a GPU runtime system optimized for AI computations under constrained GPU memory by utilizing CPU RAM offloading. TURNIP addresses the challenge of nondeterministic execution by introducing a dependency graph -- MEMGRAPH, which allows for flexible operation sequencing. The approach dynamically decides the best operation order at runtime, minimizing GPU stalls caused by memory transfers.

**Strengths:**

++ The paper targets important problem of addressing GPU utilization

**Weaknesses:**

-- The proposed approach may perform poorly when handling complex, dependency-heavy workloads

-- It is not clear what unique challenge in nondeterministic graph-based scheduling for AI computing

**Questions:**

While this paper applies a nondeterministic MEMGRAPH approach to AI computing, similar techniques have been extensively explored in compiler research for general-purpose workloads. What are the unique challenges of applying such memory management strategies to AI-specific workloads?

The non-deterministic, event-driven scheduling works best for tasks with loose dependencies. However, heavily dependent tasks are common in AI workloads (e.g., transformer models with layer-by-layer computations). When dependencies are heavy, it may leave GPUs idle because tasks are queued. How does the proposed approach work under such a case?

The paper does not seem to optimize GPU assignments or balance workloads effectively across GPUs, especially when some GPUs are busy while others are idle. This could lead to bottlenecks and inefficient use of resources in multi-GPU settings.

What is the time complexity of the MEMGRAPH construction process, particularly for very large TASKGRAPHS? How much overhead in this process?

---

> ### Author Response · Authors · 2024-11-21
>
> Below, we provide detailed responses to the reviewer’s concerns and questions.
> > W1: The proposed approach may perform poorly when handling complex, dependency-heavy workloads.
>
> We note that all of our experiments were performed using modern transformers (both training and inference). We would argue that these are among the most challenging AI workloads. By the time a modern transformer has been distributed across multiple GPUs, it produces a large, dependency-heavy workload. For example, the first token inference for one sequence of length 1024 on the 7B parameter LLaMA model with eight GPUs produces a MemGraph with 14525 non-input vertices, 2389 input vertices (which have no dependencies), and **56859** dependencies.
> > W2: It is not clear what unique challenge in nondeterministic graph-based scheduling for AI computing
>
> The central hypothesis of the paper is that modern AI co-processors introduce a high level of non-determinism, especially when considering CPU memory transfers, and so a dataflow-based approach that is designed to handle that non-determinism makes sense. To our knowledge, no existing, mainstream system for modern AI computing utilizes such an approach. That is our core contribution: to suggest to other system designers that they should consider a non-deterministic, dataflow-based approach. Also, to be clear, the paper is not concerned with scheduling. In fact, currently the TURNIP runtime does not employ a smart scheduling algorithm. TURNIP simply chooses any available operation to run, without any priority (that is, when a GPU is free, it executes whatever operation is mapped to that GPU and has all of its dependencies fulfilled).

---

> ### Author Response · Authors · 2024-11-21
>
> > Q1: While this paper applies a nondeterministic MEMGRAPH approach to AI computing, similar techniques have been extensively explored in compiler research for general-purpose workloads. What are the unique challenges of applying such memory management strategies to AI-specific workloads?
>
> The most unique aspect of our approach is that we compile the input TaskGraph into a memory access plan, with dependencies that ensure correctness. We are unaware of any prior work that does this. It is not as much that AI-related plans present special challenges, as much as they are particularly well-suited to this approach. The memory requirements of compute kernels are generally known beforehand. Further, the "non-deterministic", event-driven runtime is attractive because it is very difficult to develop analytic models for GPU-CPU memory transfers (GPUs share the same bus or buses, so they are competing for the same resources, and modeling is complicated by the fact there are often multiple NUMA nodes in a GPU server). It is our hope that others will read our paper and consider related approaches for building runtimes for AI systems.
> > Q2: The non-deterministic, event-driven scheduling works best for tasks with loose dependencies. However, heavily dependent tasks are common in AI workloads (e.g., transformer models with layer-by-layer computations). When dependencies are heavy, it may leave GPUs idle because tasks are queued. How does the proposed approach work under such a case?
>
> We note that all of our experiments are run on top of modern transformer models, and the results seem reasonable. Why? As long as we have a reasonable, multi-GPU decomposition of the input computation, in most cases there should be enough parallel work to do. Further, the particular attraction of the dataflow-based approach is that it can actually alleviate problems that arise when, due to unexpected events (such as slow memory transfers) execution of an operation is blocked. The runtime can pick any other available operation to run. All of this does rely on a reasonable decomposition of the input computation, and there are many ways to do this. As mentioned in the paper, approaches such as those embodied by Alpa and FlexFlow (not to be confused with FlexGen) already exist for performing the decomposition in the general case; in the case of transformers, one could decompose according to a special-purpose schema, such as Megatron-LM. As long as a reasonable decomposition is used, there should be enough parallel work to do.
> > Q3: The paper does not seem to optimize GPU assignments or balance workloads effectively across GPUs, especially when some GPUs are busy while others are idle. This could lead to bottlenecks and inefficient use of resources in multi-GPU settings.
>
> This is correct, our system does not load balance at runtime; the assumption is that the input Taskgraph represents a reasonable decomposition of the underlying AI computation to multiple GPUs (see our response above). In general, it is not too difficult to decompose an AI computation in a balanced way (for example, a tensor contraction such as a matrix multiplication, when decomposed 8 ways to match the 8 GPUs, should have each of its 8 sub-computations mapped to different GPUs). One could imagine adding dynamic load balancing to the system to react to cases when a GPU is not being used (for example, due to a large memory transfer). This could further improve performance, but in general, GPU-to-GPU transfers are so slow compared to the speed of the computation (even using NVLink), this is a non-trivial idea to implement. As it is, the primary method TURNIP uses to ensure all resources are being used is its ability to schedule any operation whose dependencies have all been satisfied on the GPU.
> > Q4: What is the time complexity of the MEMGRAPH construction process, particularly for very large TASKGRAPHS? How much overhead in this process?
>
> In the worst case, with a very dense graph, it would be O($V^{2}$) to produce the MemGraph. However, this is an offline cost. Given a model, and given a particular server (with a certain number of GPUs, and given memory sizes) compilation only happens one time. One can compile the graph once, and then use it for a large number of training iterations, aligning well with the typical workflow of many machine learning tasks.

---

> > ### Comment · Reviewer_mXjm · 2024-11-29
> >
> > Thanks for authors' response. However, compiling the input task into a memory access is a popular approach exsiting in compiler work for gerenal workloads, I am still not convinced about the unique challenges specific to AI computing.

---

> ### Author Response · Authors · 2024-12-01
>
> Thank you for your feedback. While static memory compilation is indeed a common approach in compilers for general workloads, AI computing presents unique challenges that exacerbate the memory scarcity problem. These challenges stem from the extremely large models used in AI, which place significant strain on GPU memory and generate substantially higher amounts of communication between the GPU and CPU.
>
> TURNIP is designed to handle dataflow graphs broadly and is not limited to AI-specific computations. However, it is particularly effective for AI workloads because it addresses these memory scarcity challenges caused by large models and the stochastic nature of GPU-CPU communications, as demonstrated in our experiments and discussed in prior comments.
>
> We acknowledge that TURNIP does not introduce a new scheduling algorithm. Instead, its novelty lies in applying a dataflow-based approach to AI computing—something we have not observed in existing systems. We hope that others will read our paper and consider related approaches for building runtimes for AI systems.

---

### Official Review · Reviewer_Bhrm · 2024-11-02

**Soundness:** 2
**Presentation:** 2
**Contribution:** 2
**Rating:** 5
**Confidence:** 4

**Summary:**

This paper presents Turnip, a software framework to support CPU memory offloading for training and inferencing deep neural networks. The key idea is to introduce a MemGraph, generated at the compilation phase, which is then used to maximize computation and memory transfer overlaps. Turnip was evaluated on 7B and 65B LLaMA models and compared against ZeRO-Inference. The experimental results demonstrate that Turnip reduces first-token latency during inference and accelerates LoRA-based model fine-tuning.

**Strengths:**

An interesting approach for incorporating dependency analysis during the compilation phase to optimize offloading and reloading operations. The evaluation was conducted using a state-of-the-art LLM.

**Weaknesses:**

A straightforward approach where the positions of tensors reloaded to GPUs are predefined at specific slots through a compilation process based on the data flow graph. This is fine for simple model architectures with a static execution flow but can struggle to deal with dynamic networks like MoEs where tensors/variables are activated based on the current input data?

The evaluation can be improved. For example, I would like to see a scalability evaluation by testing Turnip on various model sizes, as well as models with f16 during inference.


This may speed up simple model architecture, but is difficult to work with or achieve better performance on complex/dynamic model structures. Second, the experiment design could introduce more workloads such as 13B, 175B, 400B models and more scenarios such as half precision full parameter training and even more solutions such as ZeRO offload, seen Questions.

**Questions:**

What is the definition of the MemGraph referenced in Figure 5? Does it represent solely a data dependency graph with memory dependencies, or does it include additional information like control flow dependency?

In Figure 6, if the tensor locations on GPU devices are predetermined at the compilation stage, will different tensor sizes lead to GPU memory fragmentation?

How does TURNIP compare to TensorRT or other GPU-only solutions for inference? This information will help us understand its overhead.

ZeRO-Offload and ZeRO-Infinity adopt different strategies for allocating tensors between GPU devices and CPU memory. To provide a more comprehensive comparison, could you include GPU-only and ZeRO-Offload solutions in Figure 13?

Lastly, why was ZeRO-Infinity not configured with an NVMe as secondary storage, as shown in Figure 13? Doing so could potentially extend the input sequence length or support larger model sizes.

---

> ### Author Response · Authors · 2024-11-21
>
> Below, we provide detailed responses to the reviewer’s concerns and questions.
> > W1: A straightforward approach where the positions of tensors reloaded to GPUs are predefined at specific slots through a compilation process based on the data flow graph. This is fine for simple model architectures with a static execution flow but can struggle to deal with dynamic networks like MoEs where tensors/variables are activated based on the current input data?
>
> MOEs can be handled in a dataflow-based approach. When it is known that a branch in the graph is not going to be executed, all vertices in that portion of the graph are immediately marked as completed, without execution, and the graph is run normally. Further, the various parallel branches in the graph would be ordered sequentially in the vertex list V, so that allocations in one branch would not overlap with allocations in the other branch.
> > W2: The evaluation can be improved. For example, I would like to see a scalability evaluation by testing Turnip on various model sizes, as well as models with f16 during inference.
>
> We performed our evaluation on a 7B parameter model and a 65B parameter model, and we tested inference (which uses f16) and training (which uses f32). If the paper is accepted, we are happy to test our approach on larger models.
> > W3: This may speed up simple model architecture, but is difficult to work with or achieve better performance on complex/dynamic model structures. Second, the experiment design could introduce more workloads such as 13B, 175B, 400B models and more scenarios such as half precision full parameter training and even more solutions such as ZeRO offload.
>
> We have done some of those experiments, but we agree larger models would be compelling; See our response above. Also, if "dynamic" refers to models such as MoEs, as we wrote above, we can extend the TURNIP approach to handle that, and we can extend the approach to handle "looping" (inference in LLMs); this is not a fundamental deficiency of the approach, it is simply the result of engineering priorities on a research prototype system.

---

> ### Author Response · Authors · 2024-11-21
>
> > Q1: What is the definition of the MemGraph referenced in Figure 5? Does it represent solely a data dependency graph with memory dependencies, or does it include additional information like control flow dependency?
>
> As the reviewer suggests, the MemGraph is a data dependency graph with memory dependencies. We do not currently handle models with control flow (like MoEs).
> > Q2: In Figure 6, if the tensor locations on GPU devices are predetermined at the compilation stage, will different tensor sizes lead to GPU memory fragmentation?
>
> What we observe is that as memory becomes constrained, we have more frequent offloads. This, in turn, tends to alleviate fragmentation (a large allocation will "kick out" tensors from GPU RAM, effectively consolidating the memory where it is allocated). We have discussed implementing defragmentation (at compile time, by attempting to move around tensors) but it is unclear if this is truly necessary.
> > Q3: How does TURNIP compare to TensorRT or other GPU-only solutions for inference? This information will help us understand its overhead.
>
> As we have written in responses to other reviewers, we do not mean to propose TURNIP as an alternative to, or better than, a system such as TensorRT, PyTorch, TensorFlow, etc. These systems have hundreds or thousands of person-years of engineering; simply having a team of engineers work on implementing special kernels for attention or utilizing kernel fusion can have a massive impact on system performance, and undoubtedly all of these systems have a huge performance advantage in this regard (as we wrote for Reviewer 3 Q1, when memory is not constrained, TURNIP is about 2X slow than PyTorch). However, once memory becomes an issue, TURNIP is competitive or even faster than other alternatives, despite the fact it is a research prototype. That is the message of the paper: there is reasonable evidence that TURNIP’s "non-deterministic", dataflow-based approach has significant advantages once the computation becomes challenging due to the need to use CPU RAM. Our hope is that other system designers will consider such a dataflow-based approach as well, specifically as models continue to increase in size, but GPU RAM does not or could not keep up. This is the reason for our focus on the memory-constrained case.
> > Q4: To provide a more comprehensive comparison, could you include GPU-only and ZeRO-Offload solutions in Figure 13?
>
> We appreciate the excellent suggestion from the reviewer. We will include these comparisons if the paper gets accepted. (On the DeepSpeed website they claimed that ZeRO-Infinity is able to offload more data than ZeRO-Offload and has more effective bandwidth utilization and overlapping of computation and communication)
> > Q5: Why was ZeRO-Infinity not configured with an NVMe as secondary storage, as shown in Figure 13?
>
> Both TURNIP and ZeRO can support NVMe as the secondary storage, but NVMe is slower than CPU RAM, and we found that we were not constrained by CPU RAM in the larger problems (our servers had more than 1TB of CPU RAM). The attention computation itself would fail due to lack of GPU RAM before the CPU RAM would be exhausted. In TURNIP, this could be addressed by sharding the attention computation in more ways than the number of physical compute units, at the expense of additional RAM transfers (and slower runtimes), but it is unclear how this can be addressed in ZeRO Infinity. (As an aside, we point out that ZeRO uses Float16 computations for the forward pass for training, whereas TURNIP used F32; this likely accounts for the fact that TURNIP fails at slightly smaller problem sizes than ZeRO infinity in Figure 13. Modifying TURNIP to use lower precision during the forward pass is an example of the sort of optimization that can be non-trivial in a research prototype system).

---

> > ### Comment · Reviewer_Bhrm · 2024-11-25
> >
> > Thank you for your efforts in drafting the author's response. The response cleared up some of my concerns.
> >
> > However, I am not convinced that your approach can deal with MoE as the activation branches are unknown at compilation time, which can dynamically change according to the input. If the MemGraph was constructed and acted dynamically during runtime, the overhead is likely to be significant.

---

> ### Author Response · Authors · 2024-12-01
> **MoE Clarification**
>
> Thank you for your response and we make the following clarification for handling MoE in a dataflow-based approach.
>
> For MoE, all branches are included in the compiled compute graph, and we track the nodes corresponding to each branch. Once the gating network computations are completed, an additional processing step determines which branches will be executed based on the gating network's output. Nodes in branches that will not be executed are marked as complete, effectively skipping them in the computation.
>
> In this way, we do not dynamically construct the graph during runtime. Instead, we adaptively adjust the graph by marking unexecuted nodes, ensuring that only the selected branches are executed. This approach avoids the overhead of dynamically constructing the graph while accommodating the dynamic nature of MoE routing.

---

### Official Review · Reviewer_ygfA · 2024-11-03

**Soundness:** 2
**Presentation:** 4
**Contribution:** 3
**Rating:** 6
**Confidence:** 4

**Summary:**

The paper presents TURNIP, a GPU runtime that supports offloading data to CPU RAM to remove GPU memory restrictions in AI computations. TURNIP addresses the challenges of nondeterministic execution introduced by the slow data transfer rate between CPU RAM and GPU RAM, which harms computational efficiency. TURNIP’s main contribution is its use of a dependency graph (MEMGRAPH) that allows for overlapping execution of GPU kernel operations and memory transfers. The system dynamically chooses the best order of execution based on real-time events, thus minimizing idle times caused by memory transfer delays. TURNIP outperforms existing systems like PyTorch-based ones in constrained GPU environments, contributing to efficient memory management in AI workloads.

**Strengths:**

1. Originality: TURNIP accepts an abstracted version of a generic GPU computation. Other systems are more specifically targeted to certain categories of models, optimization algorithms, or specific tasks such as training or inference. None of the previous work considers the effect of the non-determinism of offload and reload operations on system performance, nor does it focus on the system runtime.
2. Quality: The quality of the design is demonstrated by experiments under various hardware conditions (A100 and P100), using LLAMA 7B & 65B.
3. Clarity: The details of the idea are clearly demonstrated with graphs and explanations in section 4-7.
4. Significance: TURNIP provides a generic solution to democratize AI computing, especially LLMs, which extends the functionality of older GPUs and enables large-scale AI model operations in constrained environments, making it valuable to both the AI and systems communities.

**Weaknesses:**

2 limitations that the author stated in the conclusion section of the paper, which potentially harms the significance of contribution (1) and soundness (2):
1. The biggest limitation of TURNIP is that the input computation (in the form of a TASKGRAPH) must be static and known completely beforehand so that the MEMGRAPH can be constructed. This is not of consequence during model training, but can be an issue during any recursive, generative AI computation. This includes LLM inference, where the next token must repeatedly be generated and the KV -cache increases in size. There are some naive solutions to this (such as pre-compiling a MEMGRAPH for a specific number of token generations in the case of an LLM) but more work will be necessary to create a satisfactory solution to recursive
computation.
2. Another limitation is that while the experiments showed that TURNIP has certain performance advantages, it is impossible to be sure where those advantages come from. Our ablation shows that a fixed execution order slows TURNIP, suggesting that non-determinism is important. But unlike ZeRO and FlexGen, TURNIP was implemented from the ground up and does not rely on
Python or PyTorch—that fact alone could account for some of the performance differences.

Others:
1. The experiments conducted in this paper use the LLAMA 1st generation model, while the newest one up to date is already LLAMA 3.2. The paper will benefit from using the SOTA LLMs to demonstrate the effectiveness of TURNIP.
2. As nowadays, the sequence length has grown rapidly (e.g., LLAMA 3.1 models have already supported 128K context window), the paper only did experiments on max 16K, so experiments on longer sequence length will be appreciated to demonstrate the effectiveness of TURNIP.
3. more diverse hardware, such as V100 and H100, can be used in the experiments if possible. Nowadays, people rarely use P100, while some still stick with V100. Meanwhile, H100 represents cutting-edge technology.
4. We already have the 405B model in LLAMA 3.1, so seeing how it performs with TURNIP can make the results more convincing.
5. In Figure 10: Time for LLaMA first token (prefill) inference, A100 server, TURNIP seems to show some scalability issues (e.g., 7B inference batch size 8, TURNIP starts to demonstrate the worst run time when the sequence length is 16K, compared to FlexGen and ZeRO Inference). Although the author mentioned that FlexGen uses very low precision arithmetic to save RAM and speed computing, this doesn't explain the issue here.

A minor issue:
1. Figures 8 & 9 are actually not figures but details of algorithms. You should call them Algorithms 1 & 2.

**Questions:**

The reviewer is happy to raise the rating if some/all the experiment issues stated are addressed.

1. Do you already have any plan to address your paper's known limitations? Since you build a customized framework that doesn't involve Python or PyTorch, For the second one, I believe a separate experiment, when no CPU offloading is involved, comparing your framework with Pytorch on top of the existing experiment parameters can somewhat contribute to the understanding of the performance differences.
2. Do you have any comments on the experiments that I think are missing? (In the weakness section, Others, 1-4)
3. Do you have any explanation for the 5th point in the weakness section under the others sub-section?
4. Just out of curiosity, would it be possible to apply TURNIP to generic graphics or data processing? I feel like it's highly possible.
5. Last question that doesn't contribute to the paper's rating: Will you open-source the code once the paper gets accepted in the future?

---

> ### Author Response · Authors · 2024-11-21
>
> Below, we provide detailed responses to the reviewer’s concerns and questions.
> > W1: The biggest limitation of TURNIP is that the input computation (in the form of a TASKGRAPH) must be static and known completely beforehand so that the MEMGRAPH can be constructed. This is not of consequence during model training, but can be an issue during any recursive, generative AI computation.
>
> This is true. However, we do think that even with this weakness, we have presented significant evidence that ML system developers should consider asynchronous, dataflow-based engines (like TURNIP) as a viable alternative, especially for a memory-constrained environment. Additional research will be necessary to completely solve problems such as this.
> > W2: But unlike ZeRO and FlexGen, TURNIP was implemented from the ground up and does not rely on Python or PyTorch—that fact alone could account for some of the performance differences.
>
> This is accurate. However, it is, unfortunately, unavoidable. The MemGraph must be run on top of an asynchronous dataflow engine---the point of the paper is that asynchronous, "non-deterministic" runtime has advantages when performing CPU offload. Thus, it is simply not possible to do an "apples to apples" comparison, as PyTorch-based systems do not implement such a runtime. (Also, see our response to Q1 below; for “easy” workloads with no offloading, TURNIP is slower than PyTorch, so it seems that the performance benefits do not come only from our use of C++).
> > Others: The reviewer suggested experiments on the newest LLaMA with longer sequence length and modern hardware
>
> We use the first generation LLaMA model. However, it is "just a workload" for TURNIP and the other systems. The differences between the various generations are quite insignificant from a systems point-of-view. Even if the newer models give much better qualitative performance, one would not expect any differences in *runtime* performance characteristics, regardless of what modern transformer is used (as long as true, all-to-all attention is run). The one exception to this would be the 405B parameter model, simply because it is so much larger. Unfortunately, this larger model was released in late July, 2024, so it was difficult to port this to Turnip in time for the ICLR deadline. With respect to the longer context size, we did run experiments with a 32K context (see Figure 12), using FP16 floats. With respect to other hardware, we agree with the reviewer that H100 GPUs would be interesting. Using H100 GPUs with 80GB of RAM, we could extend this to a 128K context---we would love to do that---but it is a problem to obtain H100 GPUs in an academic laboratory. Even the 40GB A100 GPUs that we ran experiments on are quite difficult to procure from Amazon EC2. We do have access to a V100 server and can run experiments on that, and include them in the paper, if the paper is accepted.
> > Q1: I believe a separate experiment, when no CPU offloading is involved, comparing your framework with Pytorch on top of the existing experiment parameters can somewhat contribute to the understanding of the performance differences.
>
> Testing TURNIP with no offload is a good suggestion as it can test whether PyTorch is simply slower than TURNIP, and we tried this. It turns out that TURNIP is slower than generic PyTorch for simple inference tasks when there is no offloading. For example, on the 7B parameter LLaMA model (V100 machine), on one GPU, interference takes **995ms** with TURNIP (1K sequence length) and **595ms** with vanilla PyTorch (after warm up). For the 65B parameter model TURNIP takes **1082ms** (8GPUs) and vanilla PyTorch takes **432ms**. This suggests that for simple inference tasks, PyTorch is faster than TURNIP (a lot more engineering has gone into PyTorch---it is not a research prototype like TURNIP---and developing an ML system is a difficult and time-consuming task). The fact that TURNIP is faster for memory-constrained tasks compared to these PyTorch-based systems would seem to indicate the benefit of the approach, given that it starts at a performance deficit compared to PyTorch in the simplest case.
> > Q4: Just out of curiosity, would it be possible to apply TURNIP to generic graphics or data processing? I feel like it's highly possible.
>
> Yes, TURNIP can be used for any other GPU-based application.

---

> > ### Comment · Reviewer_ygfA · 2024-11-21
> >
> > Some more comments about unresolved doubts related to my review questions also some response:
> >
> > > We use the first generation LLaMA model. However, it is "just a workload" for TURNIP and the other systems. The differences between the various generations are quite insignificant from a systems point-of-view. Even if the newer models give much better qualitative performance, one would not expect any differences in runtime performance characteristics, regardless of what modern transformer is used (as long as true, all-to-all attention is run).
> >
> > I don't think this is true. For example, at least Llama 2 (https://arxiv.org/pdf/2307.09288) introduces Grouped Query Attention (GQA, https://arxiv.org/abs/2305.13245), which is a significant architectural change from Llama 1. GQA reduces the number of query heads while keeping the number of key and value heads the same, reducing the computational complexity of the attention mechanism. By decreasing the number of query heads, GQA reduces the amount of computation and memory bandwidth required during attention calculations. The reduction in computational workload directly impacts runtime performance characteristics. Models utilizing GQA can potentially run faster and be more memory-efficient than those using standard multi-head attention. Systems executing models with GQA may exhibit lower GPU utilization, and memory consumption compared to running models without GQA. GQA changes the dynamics of the attention mechanism. While it still performs attention across all tokens, grouping query heads means that the computation pattern and resource requirements differ from standard all-to-all attention. As a result, from a systems perspective, the workload characteristics (e.g., compute intensity, memory access patterns) should be altered by architectural changes like GQA.
> >
> > Another point is that, Meta seems to no longer offer the checkpoints of their first generation of Llama, which makes reproducing challenging: https://www.llama.com/llama-downloads/
> >
> > > The one exception to this would be the 450B parameter model, simply because it is so much larger. Unfortunately, this larger model was released in late July, 2024, so it was difficult to port this to Turnip in time for the ICLR deadline. With respect to the longer context size, we did run experiments with a 32K context (see Figure 12), using FP16 floats. With respect to other hardware, we agree with the reviewer that H100 GPUs would be interesting. Using H100 GPUs with 80GB of RAM, we could extend this to a 128K context---we would love to do that---but it is a problem to obtain H100 GPUs in an academic laboratory. Even the 40GB A100 GPUs that we ran experiments on are quite difficult to procure from Amazon EC2. We do have access to a V100 server and can run experiments on that, and include them in the paper, if the paper is accepted.
> >
> > Typo: should be "405B". It's fully understandable if you don't have enough resources for the experiments, but I would appreciate it if you included the V100 experiments in your paper!
> >
> > > Testing TURNIP with no offload is a good suggestion as it can test whether PyTorch is simply slower than TURNIP, and we tried this. It turns out that TURNIP is slower than generic PyTorch for simple inference tasks when there is no offloading. For example, on the 7B parameter LLaMA model (V100 machine), on one GPU, interference takes 995ms with TURNIP (1K sequence length) and 595ms with vanilla PyTorch (after warm up). For the 65B parameter model TURNIP takes 1082ms (8GPUs) and vanilla PyTorch takes 432ms. This suggests that for simple inference tasks, PyTorch is faster than TURNIP (a lot more engineering has gone into PyTorch---it is not a research prototype like TURNIP---and developing an ML system is a difficult and time-consuming task). The fact that TURNIP is faster for memory-constrained tasks compared to these PyTorch-based systems would seem to indicate the benefit of the approach, given that it starts at a performance deficit compared to PyTorch in the simplest case.
> >
> > Good to know these, these statistics can be included in your paper. Regarding the statement in your paper, "But unlike ZeRO and FlexGen, TURNIP was implemented from the ground up and does not rely on Python or PyTorch—that fact alone could account for some of the performance differences.", you can make it clear and say that your custom implementation roughly doubles the latency of Pytorch.
> >
> > > In Figure 10: Time for LLaMA first token (prefill) inference, A100 server, TURNIP seems to show some scalability issues (e.g., 7B inference batch size 8, TURNIP starts to demonstrate the worst run time when the sequence length is 16K, compared to FlexGen and ZeRO Inference). Although the author mentioned that FlexGen uses very low precision arithmetic to save RAM and speed computing, this doesn't explain the issue here.
> >
> > By the way, the authors haven't had a response on my W5, which is my Q3.

---

> ### Author Response · Authors · 2024-11-22
> **Response to W5**
>
> Thank you for your constructive feedback and we apologize for missing the response to W5 in our initial response.
> > In Figure 10: Time for LLaMA first token (prefill) inference, A100 server, TURNIP seems to show some scalability issues (e.g., 7B inference batch size 8, TURNIP starts to demonstrate the worst run time when the sequence length is 16K, compared to FlexGen and ZeRO Inference).
>
> While acknowledging the complexity of these systems, which makes it challenging to fully understand every detail, we are confident that we can explain the majority of these results.
>
> One notable result is that FlexGen is by far the slowest for the shortest sequence lengths, but the performance decreases slowly as the problem gets harder. The slow response time for FlexGen is due to the fact that it is the only of the three systems to use pipelined parallelism, which does little to lower latency. But FlexGen’s use of reduced precision helps flatten the curve, as it pays a lower penalty for both compute and potential GPU offload than the other two systems. This is why FlexGen scales well with larger problems, though it is relatively slow overall, especially on the smaller problems.
>
> ZeRO and TURNIP scale quite similarly, though TURNIP is a constant factor faster. The case where this relationship breaks down a bit is on the larger problems (the most obvious case is for the 7B LLaMA model, size 8 batch, 16K sequence, A100 server, where TURNIP is significantly slower). On these larger problems is where some of the difficulties associated with running a research-prototype system really start to matter. In particular, in the case described above, about 90% of the RAM on each GPU will be needed just to perform the softmax attention operation in TURNIP when it is decomposed 8 ways, and so this operation empties all other data from RAM. There are obvious ways to address this, but optimizing attention is largely orthogonal to the core topic of the paper.

---

> > ### Comment · Reviewer_ygfA · 2024-11-23
> >
> > Thank you for your response! Some comments:
> >
> > > One notable result is that FlexGen is by far the slowest for the shortest sequence lengths, but the performance decreases slowly as the problem gets harder. The slow response time for FlexGen is due to the fact that it is the only of the three systems to use pipelined parallelism, which does little to lower latency. But FlexGen’s use of reduced precision helps flatten the curve, as it pays a lower penalty for both compute and potential GPU offload than the other two systems. This is why FlexGen scales well with larger problems, though it is relatively slow overall, especially on the smaller problems.
> >
> > This is fully understandable, the authors already have noted these in the paper, and my main focus on those figures is not about FlexGen.
> >
> > > ZeRO and TURNIP scale quite similarly, though TURNIP is a constant factor faster. The case where this relationship breaks down a bit is on the larger problems (the most obvious case is for the 7B LLaMA model, size 8 batch, 16K sequence, A100 server, where TURNIP is significantly slower). On these larger problems is where some of the difficulties associated with running a research-prototype system really start to matter. In particular, in the case described above, about 90% of the RAM on each GPU will be needed just to perform the softmax attention operation in TURNIP when it is decomposed 8 ways, and so this operation empties all other data from RAM. There are obvious ways to address this, but optimizing attention is largely orthogonal to the core topic of the paper.
> >
> > This part is not noted in the paper. You should add these in your manuscript to make the potential issue clear. Although you mentioned that optimizing attention is largely orthogonal to the core topic of the paper, to testify your arguments and also to ***demonstrate the effectiveness of TURNIP on SOTA AI workloads***, SOTA attention memory optimizing techniques should, of course, be used for your experiments, which includes the newer generation of Llama models (i.e. because of the use of GQA as previously mentioned), as well as flash attention (https://arxiv.org/abs/2205.14135) or something similar (such as xFormers https://facebookresearch.github.io/xformers/). I assume that you didn't use them, as I don't see them in your code or paper.
> >
> > > With respect to the longer context size, we did run experiments with a 32K context (see Figure 12), using FP16 floats.
> >
> > In addition, as a comment to the author's initial response on the context length issue (which I forgot to mention in my last response), although in Figure 12, the authors run experiments with a 32K context, this is not the case in Figure 10. Also, as there is no OOM demonstrated in time for LLaMA first token (prefill) inference, 8 x A100 server related to the 7B LLaMA model, batch size 8 for up to 16K context, and batch size 16 for up to 8K context (first two subgraphs of Figure 10), longer sequence length should, of course, be experimented until TURNIP or ZeRO or FlexGen have OOM, so that we can get the whole picture.
> >
> > Unfortunately, as the ICLR public discussion phase is ending and it's not likely that these experiments can be finished before the deadline, I will still stick to 6 for now. To make it clear again, I might raise it to 8 if the following experiments are done and they can ***demonstrate the effectiveness of TURNIP on existing AI workloads***:
> > 1. Switch to newer generation of Llama models, minimum Llama 2.
> > 2. Use fast and memory-efficient attention-optimizing techniques for your experiments, such as flash attention or xFormers.
> > 3. Longer context length in the first two subgraphs of Figure 10 until one has OOM.
> >
> > It would, of course, be a plus point if modern hardware and larger model parameter sizes could be used in the experiments, but this is totally optional given the fact that the authors have limited access to resources.

---

### Official Review · Reviewer_P5t5 · 2024-11-04

**Soundness:** 2
**Presentation:** 2
**Contribution:** 2
**Rating:** 5
**Confidence:** 3

**Summary:**

This paper presents a runtime system for executing AI applications via GPU-to-CPU offloading. The system dynamically determines whether to allocate AI computations to GPU or CPU memory. It begins by transforming a task graph into a memgraph and implementing the memgraph engine through simulations. This approach enables efficient execution of AI computations through best order of actions (offloading or reloading).

**Strengths:**

Provided detailed explanations of how to generate memgraph

Implemented a working system

**Weaknesses:**

The paper lacks detailed discussions on how Turnip differs from existing systems mentioned in the related work section.

It is unclear how the simulation for memgraph is generated.

The simulation-based memgraph implementation raises questions on how the system guarantees optimal action sequencing during runtime.

Has the system been opensourced?

There are language issues here and there:
Page 3: Turnip takes [as] input ...
Three GPUs, each [having] five ...
With seven tensors [in] total ...

**Questions:**

How is the simulation obtained to generate the memgraph?

How does the system ensure that operations are executed in the best order at runtime?

Has the system been open-sourced?

---

> ### Author Response · Authors · 2024-11-21
>
> Below, we provide detailed responses to the reviewer’s concerns and questions.
> > W1: The paper lacks detailed discussions on how Turnip differs from existing systems mentioned in the related work section.
>
> TURNIP is based upon the idea of using an asynchronous dataflow engine to power ML computations. As we wrote to Reviewer 1 W1, no major modern ML system is based on an asynchronous dataflow engine (a "modern" ML system might be defined as a system that can execute a modern LLM), and such proposals are absent from the recent ML literature. That is the main point of our work: there may be significant advantages to using such an engine, especially with respect to memory management. That is how our work differs from all other work in this space.
> > W2: It is unclear how the simulation for memgraph is generated.
>
> If the reviewer is asking about how the vertex order V is generated, this is a topological sort of the vertices, with some optimizations (see our response to Reviewer 1 W2). If the reviewer is asking how the actual compilation happens, the algorithms in the paper cover that in depth. We will provide additional implementation details in an Appendix. We have a simulated allocator, and we have a set of self-defined allocation functions, similar to malloc, but with some extra information that stores memory dependencies (a tensor can only be placed at a certain memory location after another tensor has been deleted). We also have a set of states that keeps track of the location of each tensor while it's active. In every iteration we will try to simulate allocation of an output-tensor on the GPU RAM. If the allocator is full, then we will trigger the offload/reload algorithm in order to find a space.
> > W3: The simulation-based memgraph implementation raises questions on how the system guarantees optimal action sequencing during runtime.
>
> As we wrote in our response to Reviewer 1 W2, the MemGraph is a dependency graph so it does not force any execution order, and the actual execution order at runtime could differ from run to run, depending on the real-time ordering of events. The choice of V does have an impact, however, as it influences the set of memory dependencies that are added. The memory dependencies do constrain the available execution orderings at runtime so that it is impossible to have an execution order that does not produce a correct result. One of the major points of the paper is that given the asynchronous and seemingly stochastic nature of memory transfers, it is difficult/impossible to plan an "optimal" order beforehand. Instead, TURNIP attempts to build a MemGraph whose dependencies constrain runtime actions as little as possible. That way, TURNIP can choose to run any operation that is available, whenever a GPU is free, and the chance of being blocked is minimized.

---

> > ### Comment · Reviewer_P5t5 · 2024-11-25
> >
> > Thanks authors for their response to my comments. However, the authors' response does not address my concern regarding memgraph. There appears to be no mechanism to ensure the efficient execution order of the operations, based on the memgraph generated by simulations.  The method guaranttees the validity of the operation order, not its quality or efficiency.

---

> ### Author Response · Authors · 2024-12-01
>
> Thank you for your response. We would like to clarify further with regard to the execution order.
>
> >The method guaranttees the validity of the operation order, not its quality or efficiency.
>
> It is correct that TURNIP does not provide a theoretical guarantee regarding the quality or efficiency of the execution order, and we will clarify this further in the limitation section of the paper. However, due to the stochastic nature of memory transfers, we believe no analytic cost model can precisely predict execution order, meaning no compile-time ordering can consistently achieve optimal performance.
> TURNIP operates by executing computations as soon as they become available, constrained only by the dependencies in the MemGraph. Empirically, this approach has demonstrated better performance compared to any pre-determined forced ordering of the nodes.

---

### Official Review · Reviewer_t5jx · 2024-11-08

**Soundness:** 2
**Presentation:** 3
**Contribution:** 2
**Rating:** 5
**Confidence:** 3

**Summary:**

This paper presents TURNIP, a GPU Runtime that incorporates CPU-RAM offload into the execution graph of AI models under memory constraints. Specifically, the authors create a MemGraph and GPU memory mapping by simulating task graphs. Comparing with PyTorch-based systems, they show faster runtimes of LLMs primarily for short sequence lengths.

**Strengths:**

- The paper is well-written and the examples are helpful
- The problem addressed in this work is interesting and relevant
- Using LLaMA and LoRA in the experiments makes this work applicable

**Weaknesses:**

- The idea of "nondeterministic" is not new. The described scheduler is simply a work-conserving dynamic scheduler.
- The simulation of the task graph to create the MemGraph is omitting any possible knowledge of execution times. Therefore, the resulting memgraph is only dependent on the topological sort and thus likely suboptimal.
- The authors only present memory allocation sizes of 1 unit each and claim that "In the 'real life' case where tensors are variably-sized, the algorithm does not change appreciably". Intuitively, the problem should get significantly harder with variable size allocations. The authors should elaborate on this.
- Section 7 could benefit from a more formal analysis.
- In Section 8: "Note that these are PyTorch-based systems, whereas TURNIP is not." Does that mean the authors are running their system in C++ while the others are running in Python? Does this explain the initial performance benefit for low sequence lengths? They should run a test of a model that has no memory problems. In that test all algorithms should be the same.
- Section 8, Nondeterministic vs deterministic: In the deterministic case precedence constraints are added based on a topological order which has many solutions. Therefore, it is unsurprisingly weaker. The authors should compare to any other scheduling algorithm, such as priority-based algorithms (e.g. priority-based list scheduling).

Minor issues:
- l147 takes is -> takes as
- Pacement of figures in Sec 4 is not ideal. Try placing them where they are discussed.
- Fig. 7: Why is there a memory dependency from 4 to 3. It should at least be dashed.
- Figure 8 -> Algorithm 1, Figure 9 -> Algorithm 2
- Figure 8: Assignment and comparison of v_allocHzn.loc in the first "if" clause is difficult to understand

**Questions:**

1) What is the effect of C++ vs Python?
2) Please elaborate on the effects of variable size allocations on the Memgraph generation
3) How does it compare to other existing work-conserving, priority-based algorithms?

---

> ### Author Response · Authors · 2024-11-21
>
> Below, we provide detailed responses to the reviewer’s concerns and questions.
> > W1: The idea of "nondeterministic" is not new. The described scheduler is simply a work-conserving dynamic scheduler.
>
> We definitely agree---proposals for the type of dataflow engine implemented by TURNIP date back at least to Karp and Miller (1966). In retrospect, this should have been made clearer in the introduction/related work, and we will do this. However, no major modern ML system is based on an asynchronous dataflow engine (a "modern" ML system might be defined as a system that can execute a modern LLM), and such proposals are absent from the recent ML literature. That is the main point of our work: that there may be significant advantages to using such an engine, especially with respect to memory management (a secondary, more technical contribution of the paper is the idea of compiling a dataflow graph into a "MemGraph" that is free from race conditions, by design, and requires no expensive dynamic memory allocation). Our hope is that TURNIP will inspire other ML system designers to consider an asynchronous/"non-deterministic" approach.
> > W2: The simulation of the task graph to create the MemGraph is omitting any possible knowledge of execution times. Therefore, the resulting memgraph is only dependent on the topological sort and thus likely suboptimal.
>
> The simulation itself simply accepts the ordering of vertices V provided, though it can work with any ordering that respects the dependencies TaskGraph dependencies. The reviewer is correct that this ordering can affect the quality of the resulting dataflow graph. In fact, our implementation of TURNIP uses its awareness of the underlying operations to optimize the ordering. For example, all of the kernel calls that make up a decomposed contraction (which will have no dependencies between each other and should run in parallel) are provided one-after-another in V. We will add a section in the Appendix that addresses this issue, as well as adding an ablation study to determine the importance of optimizing V. One other clarification: as the MemGraph is a dependency graph, it does not force any execution order, and the actual execution order at runtime may not match V. The choice of V does have an impact, however, as it influences the set of memory dependencies that are added. With sufficiently limited memory, the memory dependencies will constrain the available execution orderings at runtime–but only with limited memory. When the memory is sufficiently large, there will be no memory dependencies and the TaskGraph and MemGraph will coincide.
> > W3: Intuitively, the problem should get significantly harder with variable size allocations.
>
> Extending to variable-sized allocation blocks is not difficult. We would modify the simMalloc algorithm around line 330 as follows: Instead of searching for a single slot that has never been allocated or has been allocated far into the past, search for a range or interval of slots to minimize the latest dependency. This is what we do in our implementation. We agree with the reviewer that this would be useful to include in the paper, and if it is accepted, we can add an Appendix with this full algorithm.
> > W4: Section 7 could benefit from a more formal analysis.
>
> Thank you for your helpful suggestion. We will turn these observations into proper theorems.

---

> ### Author Response · Authors · 2024-11-21
>
> > W5: Does that mean the authors are running their system in C++ while the others are running in Python? Does this explain the initial performance benefit for low sequence lengths?
>
> TURNIP was built from scratch using C++, rather than on top of PyTorch. This is necessary as the core TURNIP engine is so different from PyTorch or other existing ML systems. However, this should not be taken to imply that these systems have a disadvantage due to their use of Python. PyTorch uses a lightweight Python wrapper that sits on top of a large, C++ code base. We will clarify this. (Also, see our response to Reviewer 3 Q1; for “easy” workloads with no offloading, TURNIP is about twice as slow as PyTorch, so it seems that the performance benefits do not come only from our use of C++).
> > W6: The authors should compare to any other scheduling algorithm, such as priority-based algorithms (e.g. priority-based list scheduling).
>
> Priority-based list scheduling is a dynamic scheduling algorithm, so it is "non-deterministic" as well. The point of the paper---which we will try to make clearer based on the review---is not that dataflow-based or dynamic scheduling is new. We certainly did not invent it. The point is that ML systems do not use these ideas. The core hypothesis of the paper is that existing ML systems need to make use of these classic ideas, and if they do, performance will improve (especially in memory-constrained environments where it is very difficult to statically plan for loads and offloads). Also, TURNIP runtime could be modified to accept priorities, and hence it could be modified to implement a priority-based scheduler. In fact, this is an excellent direction for future work---it would likely improve TURNIP. However, doing this may require solving an interesting research problem: how to prioritize operations to minimize runtime?

---

### Author Response · Authors · 2024-11-21
**Global Response and Code Release**

We sincerely thank the reviewers for dedicating their time to carefully reviewing our paper and providing valuable feedback. We appreciate their recognition that our work addresses a significant issue in machine learning (t5jx, ygfA, mXjm), introduces a novel approach to tackling the memory scarcity problem (ygfA, Bhrm), and evaluates its effectiveness on state-of-the-art LLMs (t5jx, Bhrm). Additionally, Reviewer P5t5 acknowledged our effort in building the system. We appreciate the thoughtful constructive comments as well.

We do have one global clarification for all reviewers with respect to the central contribution of our work, which is an issue that came up in several of the reviews. The central contribution of the paper is to show that a “non-deterministic” dataflow-based system with pre-planned memory usage and offloads can have significant performance advantages when GPU memory is constrained. This is a novel claim, as no mainstream AI system utilizes a similar strategy. Because such a runtime is so different from what systems such as PyTorch implement, implementing TURNIP from the ground up was the best way to investigate our ideas in a reasonable timeframe. Ultimately, we hope that our work inspires other system designers (such as those maintaining PyTorch) to consider incorporating our strategies into their own systems.

For transparency and reproducibility, we have released our code via an anonymous GitHub repository: https://anonymous.4open.science/r/einsummable-7CDF/.

Below, we provide detailed responses to the reviewers’ questions and comments. If there are any further concerns or additional questions, please feel free to reach out.

---

### Meta-Review · Area_Chair_WrVo · 2024-12-20

**Metareview:**

Summary
The paper proposes a framework to make the inference of large neural networks more memory efficient. The proposed framework, called Turnip, offloads CPU memory by creating MemGraph which is a dependency graph created at compilation time and MemGraph can be used to decide the best operations at runtime which reduces GPU idle times triggered primarily by memory transfers.

Strengths
1. Improving inference resources needed for large models is very important.
2. The method is tested on Llama 7B and 65B models and compared against ZeRO inference. The key result is that the latency for first token generation is reduced. The authors study this on different hardware too.
3. Since inference is improved, the authors also demonstrate speed ups for LoRA based training.

Weaknesses
1. The biggest limitation is the requirement of a static computation graph. This is a strict requirement and not true for most modern LLMs, especially MOE based models. This severely limits the impact of this paper.
2. The novelty of this work is limited as it can be viewed as compiling the model's execution into a memory access plan. While this is effective, it is a widely used technique in systems across computer science.
3. Limitations of the evaluation setup in terms of model sizes, floating point precisions (fp8)

Justification
The paper received a majority negative review primarily because of its limitation using static compute graphs and limited novelty. The AC agrees with this majority recommendation from the reviewers.

**Additional Comments On Reviewer Discussion:**

The authors tried to address the reviewers concerns. However, even after the rebuttal, a majority of the reviewers remain unconvinced about the contribution of this paper.

---

### Decision · Program_Chairs · 2025-01-22

Reject